# Palliation in Gallbladder Cancer: The Role of Gastrointestinal Endoscopy

**DOI:** 10.3390/cancers14071686

**Published:** 2022-03-26

**Authors:** Tommaso Schepis, Ivo Boškoski, Andrea Tringali, Vincenzo Bove, Guido Costamagna

**Affiliations:** 1Digestive Endoscopy Unit, Fondazione Policlinico Universitario Agostino Gemelli IRCCS, Largo A. Gemelli, 00168 Rome, Italy; tommaso.schepis@gmail.com (T.S.); andrea.tringali@unicatt.it (A.T.); vincenzo.bove@policlinicogemelli.it (V.B.); guido.costamagna@unicatt.it (G.C.); 2Centre for Endoscopic Research, Therapeutics and Training (CERTT), Università Cattolica del Sacro Cuore di Roma, Largo F. Vito, 00168 Rome, Italy

**Keywords:** gallbladder cancer, endoscopy, endoscopic palliation

## Abstract

**Simple Summary:**

Gallbladder cancer is burdened by poor prognosis, and palliation often represents the best option. Endoscopy plays a crucial role in the management of cholangitis, jaundice, gastric outlet obstruction, and pain. The aim of this study is to perform a review of the literature to assess the role of endoscopy in the palliative therapy of advanced gallbladder malignancy.

**Abstract:**

Gallbladder cancer is a rare malignancy burdened by poor prognosis with an estimated 5-year survival of 5% to 13% due to late presentation, early infiltration of surrounding tissues, and lack of successful treatments. The only curative approach is surgery; however, more than 50% of cases are unresectable at the time of diagnosis. Endoscopy represents, together with surgery and chemotherapy, an available palliative option in advanced gallbladder cancers not eligible for curative treatments. Cholangitis, jaundice, gastric outlet obstruction, and pain are common complications of advanced gallbladder cancer that may need endoscopic management in order to improve the overall survival and the patients’ quality of life. Endoscopic biliary drainage is frequently performed to manage cholangitis and jaundice. ERCP is generally the preferred technique allowing the placement of a plastic stent or a self-expandable metal stent depending on the singular clinical case. EUS-guided biliary drainage is an available alternative for patients not amenable to ERCP drainage (e.g., altered anatomy). Gastric outlet obstruction is another rare complication of gallbladder malignancy growing in contact with the duodenal wall and causing its compression. Endoscopy is a less invasive alternative to surgery, offering different options such as an intraluminal self-expandable metal stent or EUS-guided gastroenteroanastomosis. Abdominal pain associated with cancer progression is generally managed with medical treatments; however, for incoercible pain, EUS-guided celiac plexus neurolysis has been described as an effective and safe treatment. Locoregional treatments, such as radiofrequency ablation (RFA), photodynamic therapy (PDT), and intraluminal brachytherapy (IBT), have been described in the control of disease progression; however, their role in daily clinical practice has not been established yet. The aim of this study is to perform a review of the literature in order to assess the role of endoscopy and the available techniques in the palliative therapy of advanced gallbladder malignancy.

## 1. Introduction

Gallbladder cancer (GBC) is a rare malignancy accounting for 1.2% of all cancers worldwide and representing 50% of biliary tract malignancies [1]. The incidence of GBC varies depending on the geographic regions, with higher incidence reported in South America and Asia [2]. Female sex, age > 60 years, obesity, family history, gallstones (particularly those with size > 3 cm), porcelain gallbladder, gallbladder polyps, anomalous pancreaticobiliary junction, chronic biliary tract infection (e.g., Salmonella Typhi), primary sclerosing cholangitis, and smoking are considered risk factors for GBC [3]. The majority of GBC derives from the epithelial surface (98%), with adenocarcinoma representing the most common histotype (76%–90% of cases), followed by papillary, squamous, and adeno-squamous lesions [4]. Rarely, GBC can arise from non-epithelial tissues with the occurrence of lymphomas, sarcomas, and neuroendocrine tumors [5]. Depending on the regional and distant spreading, GBC can be classified in stage I (when the tumor is confined only in the gallbladder -T1 N0 M0-), stage II (when the tumor has extended to the perimuscular connective tissue -T2 N0 M0-), stage IIIA (when the tumor has spread beyond the gallbladder but not contacting arteries or veins -T3 N0 M0-), stage IIIB (when a tumor of any size has spread to lymph nodes -T1,2,3 N1 M0-), stage IVA (when the tumor has invaded the main portal vein or hepatic artery and/or lymph nodes -T4 N1 M0-), and stage IVB (tumor of any size with distant metastasis or distant lymph nodes involvement -any T N2 or M1-) [6]. The only curative approach for GBC is surgery. It requires a simple cholecystectomy in the early stages or a major hepatectomy and lymphadenectomy in more advanced stages [7]. However, more than 50% of cases are unresectable at the time of diagnosis, and indeed, GBC is burdened by poor prognosis with an estimated 5-year survival of 5% to 13% due to late presentation, early infiltration of surrounding tissues, and lack of successful treatments [8]. The lack of muscularis mucosae and submucosa and the direct venous drainage from the gallbladder to the liver facilitate the spread of GBC to adjacent structures [6]. Moreover, the dissemination can also occur via intraductal spread through the hematogenous, lymphatic, neural, and peritoneal routes [9].

GBC is generally asymptomatic at early stages, and it is frequently diagnosed accidentally after cholecystectomy for lithiasic or inflammatory diseases or during abdominal imaging performed for other reasons [3]. When present, symptoms are non-specific, ranging from dyspepsia to abdominal pain. Laboratory tests may show abnormal liver function, cholestasis, and increased oncomarkers (e.g., CA19-9 and CEA) [10]. In advanced GBC, the involvement of adjacent structures can lead to severe symptoms, significantly reducing the patient’s quality of life. The involvement of the main biliary duct can determine the occurrence of jaundice and cholangitis [11]. Moreover, the close anatomical connection of the gallbladder with the gastrointestinal tube can lead to gastric or intestinal obstruction [12]. Finally, the pathological involvement of neurological structures such as the celiac plexus can be responsible for severe abdominal pain requiring medical or surgical management [13].

Endoscopy, together with surgery and chemotherapy, represents an available palliative option in advanced GBC not eligible for curative treatments. The aim of this paper is to perform a review of the literature to assess the role of the endoscopic palliative management of advanced GBC presenting with locoregional complications (e.g., jaundice, cholangitis, gastric outlet obstruction, or incoercible pain) and the role of endoscopic locoregional therapy to control cancer progression.

## 2. Jaundice and Cholangitis

A total of 33% to 56% of patients with GBC present jaundice at the diagnosis [14]. Jaundice represents a poor prognostic factor and is frequently associated with an advanced GBC not suitable for curative treatments [15]. Jaundice in GBC can be determined by cancer intraductal spreading, direct infiltration of the common bile duct or the hilum, compression from lymph nodal metastasis, incidental biliary stone, or extensive liver metastasis [16,17]. More rarely, the patient can develop cholangitis, a life-threatening condition determined by an infectious process within the biliary tree [18]. Cholangitis, incoercible pruritus, jaundice with abdominal pain, and deep jaundice are considered indications for biliary drainage in order to reduce the risk for recurrent cholangitis, improve symptoms (e.g., pruritus and anorexia) and the quality of life [19].

Endoscopic biliary drainage (EBD), percutaneous transhepatic biliary drainage (PTBD), and surgical bypass are the available alternatives for jaundice palliation [20]. Surgical bypass is more invasive and is generally not preferred. EBD and PTBD are considered the first option; however, the choice between the endoscopic and the percutaneous approach is not standardized and frequently depends on local facilities. In palliative biliary drainage, EBD is generally preferred rather than PTBD, which is associated with several complications, including stent occlusion, complete dislodgement, cholangitis, peri-catheter leakage, bleeding, local infection, and patients’ discomfort [20]. Therefore, EBD has been reported to be associated with a lower adverse events rate and shorter hospitalization [21]. Hameed et al., in a meta-analysis of 15 studies, showed that patients undergoing PTBD presented a higher incidence of liver failure and a shorter 1-year and 5-year survival when compared with EBD [22]. Mahjoub et al. in a metanalysis involving 433 patients, reported that post-operative morbidity and mortality were higher in the PTBD group when compared with EBD (respectively 26% vs. 21% and 7.5% vs. 3.8%), although a lower incidence of procedure-related complications in the PTBD group [23].

### 2.1. ERCP-Guided Biliary Drainage

ERCP is the cornerstone of EBD, offering three different options to perform the biliary drainage: the placement of endoscopic naso-biliary drainage (ENBD), a plastic stent (PS), or a self-expandable metal stent (SEMS).

ENBD provides the insertion of a 5Fr-7Fr drainage tube inside the biliary tree and its exteriorization through the nose to allow external drainage [24]. Lin et al. in a metanalysis involving 925 patients, reported a lower rate of cholangitis, pancreatic fistula, and stent dysfunction in patients undergoing ENBD when compared with stent placement [25]. ENBD can be considered an option in pre-operative biliary drainage as a bridge to surgery. However, ENBD is inadequate for palliation considering the patient’s discomfort e the risk for tube displacement [26].

PS and SEMS can be inserted into the biliary tract to obtain internal drainage. Major advantages of PS are the removability, the moldability (the caliber and the length of PS can be adapted to the singular biliary tree shape) (Figure 1), and the inexpensiveness [27]. On the other side, PS are burdened by the risk of stent migration (occurring in 5%–10% of cases) and stent occlusion (occurring in 30% of cases because of bacterial biofilm formation, biliary sludge, biliary reflux of dietary fibers, and clots formation) [28]. Given the risk for stent occlusion, PS exchange is generally needed every 3–6 months [29].

SEMS are considered the gold standard for the EBD with palliative purpose. Therefore, SEMS are associated with a lower risk of short-term and long-term stent occlusion, lower incidence of therapeutic failure, lower cholangitis rate, fewer need for reinterventions, and longer overall survival when compared with PS [30,31]. The fully covered SEMS (FC-SEMS), the partially covered SEMS (PC-SEMS), and the uncovered-SEMS (U-SEMS) are the three available options.

The U-SEMS presents a metallic mesh that enables the complete drainage of the side biliary branches [32]. The ingrowth of the tumor inside the stent through the metallic mesh determines a close adhesion between the biliary three and the U-SEMS, thus reducing the risk for stent migration but impeding the eventual stent removal [33]. Therefore, U-SEMS are frequently used for palliative purposes when the stent is left in place indefinitely (Figure 2). Conversely, FC-SEMS presents a plastic polymer around the metallic mesh reducing the risk for tissue ingrowth and allowing an easier removal but increasing the risk for stent migration and septic complications (the plastic covering impairs the drainage of side biliary branches) [34]. PC-SEMS presents uncovered proximal and distal ends and a plastic polymer in the central portion reducing the risk for stent migration and maintaining the possibility of stent removal [35]. FC-SEMS and PC-SEMS can be used for strictures not involving the hilum and when the EBD is required before the curative surgery or when the diagnostic algorithm has not been completed yet. U-SEMS should be preferred for palliative drainage, especially when multiple stents are needed avoiding, therefore, the obstruction of side branches [32,36].

In the case of complex hilar biliary strictures, the placement of multiple SEMS may be needed to obtain complete biliary drainage. The stent-in-stent (SIS) and side-by-side (SBS) techniques are the two available options for the placement of bilateral SEMS [33]. The SIS method implies the insertion of the first SEMS over a guidewire in one emibiliary system, and subsequently the insertion of the second SEMS into the other emibiliary system via the central open mesh of the first SEMS [36]. The SBS method provides the parallel insertion of two SEMS in the two emibiliary systems [33]. The SBS method is generally preferred over the SIS for the easier deployment technique, and the easier reintervention in case of SEMS misfunction [36]. The studies comparing the two techniques are sparse, and there is no evidence demonstrating the superiority of one of the two techniques.

Dao-jian et al. performed a retrospective study comparing metal versus plastic stents for unresectable gallbladder cancer with hilar biliary obstruction in 59 patients. They reported no differences between the two groups in terms of clinical success, the occurrence of early adverse events, and later cholangitis. However, stent patency was longer in the metal stent group (119 vs. 93 days) [37]. Similarly, Tsuyoshi et al. performed a randomized control trial to compare SEMS vs. PS in 60 patients with malignant biliary stricture (in 13 patients, the etiology was gallbladder cancer) [38]. The reported 6-month patency rate was significantly higher in SEMS (81% vs. 20%), the 50% patency period was 359 days in the SEMS group, and 112 days in the PS group, the mean number of reinterventions was higher in PS groups, and the overall total costs were lower in SEMS group.

### 2.2. EUS-Guided Biliary Drainage

When the transpapillary biliary drainage is technically difficult (e.g., in gastric outlet obstruction, in tight biliary stricture, or in altered anatomy), the EUS-guided biliary drainage (EUS-BD) can represent a valid alternative to EBD [39]. With EUS, the biliary tract can be accessed from the intrahepatic or the extrahepatic route. The intrahepatic approach involves the transgastric puncture of the left hepatic system and the consequent insertion of a guidewire throughout the papilla [40]. The extrahepatic approach provides the direct puncture of the dilated common bile duct (Figure 3) [41]. Once the biliary tree has been accessed, the biliary drainage can be performed: (1) transmurally by placement of a stent in order to create a direct communication between the biliary tree and the digestive tube (the available techniques are the choledochoduodenostomy, choledochogastrostomy, hepaticoduodenostomy, and hepaticogastrostomy); (2) anterogradely by the passage of a guidewire through the stenosis until the papilla and the releasing of a stent; (3) retrogradely (rendez-vous technique) by the insertion of a guidewire until the papilla, exchange of the endoscopice with a duodenoscope and placement of a stent via the papilla using the guidewire as a route [42].

Jin et al. performed a metanalysis including 302 patients with distant malignant biliary obstruction treated with EUS- or ERCP-guided biliary drainage [43]. They reported no differences in terms of technical success rate, clinical success rate, and total adverse events between the two procedures. EUS-BD presented a lower incidence of post-procedural pancreatitis, stent dysfunction, and tumor ingrowth. In another metanalysis, Han et al. included 756 patients and reported no significant differences between EUS-BD and ERCP in terms of success rate and safety [44].

## 3. Gastric Outlet Obstruction (GOO)

Patients with advanced GBC can present mechanical gastric outlet obstruction (GOO) due to duodenal compression or infiltration [45]. The clinical presentation of GOO includes early satiety, postprandial epigastric tenderness, nausea, vomiting, anorexia, weight loss, epigastric pain, and dyselectroliteemia [46]. Most patients with GOO have a short life expectancy if left untreated and present a significant reduction in the quality of life [47]. Traditionally surgery was considered the standard palliative treatment for patients with GOO [48]. The surgical technique provides an open or laparoscopic gastrojejunostomy. Although this technique is associated with suitable clinical outcomes allowing symptoms relief in a high percentage of patients, it is an invasive procedure burdened by operative and peri-operative complications in such fragile patients [49]. Endoscopy represents a less invasive alternative to manage the GOO.

### 3.1. Endoscopic Self-Expandable Metal Stent

As described for biliary strictures, also for gastro-duodenal stenting, there are three different options: U-SEMS, PC-SEMS, and FC-SEMS [50]. PC-SEMS and FC-SEMS present a silicon or plastic membrane covering the metallic mesh that reduces the risk for tumor ingrowth but increases the risk for stent migration [51]. Tringali et al. performed a metanalysis of 9 studies, including 1741 patients with GOO treated with endoscopic placement of U-SEMS or FC-SEMS [52]. They reported a higher risk for stent occlusion in U-SEMS and a higher risk for migration in FC-SEMS. No differences were found in terms of success rate, overall adverse events, reintervention rate, dysfunction rate, and patient survival. Generally, in patients with poor prognosis and with duodenal stenosis near the major papilla, U-SEMS are considered the first option (Figure 4) [51]. Covered stents are preferred in benign strictures to allow the eventual stent removal [53]. The correct position of the SEMS across the stricture is critical in order to increase clinical success and reduce the occurrence of adverse events [54]. Various stent lengths are available for a correct adaptation to patients’ characteristics. In a systematic review by Dormann et al., the reported technical and clinical success rates of endoscopic stenting were, respectively, 97% and 97.4% [55]. Similar results were reported by Halsema E. et al. in a pooled analysis of 19 prospective studies involving 1281 patients with GOO [56]. The technical and clinical success rates were 97.3% and 85.7%, respectively. Jeurnink et al., in a systematic review, compared the outcomes of endoscopic stent placement and surgical gastrojejunostomy.

No differences were found in terms of technical success (96% vs. 100%) and adverse events occurrence (early AEs 7% vs. 6% and late AEs 18% vs. 17%) [57]. Although the suitable clinical and technical success rate, endoscopic stenting is burdened by early and late AEs. Stent occlusion is the most common complication (incidence range 8%–25.4%) due to tumor ingrowth or food impaction [58]. Stent occlusion due to tumor ingrowth can be managed with the placement of a coaxial stent (stent-in-stent technique) [59]. Stent migration is less common (incidence range 0%–19.4%) and occurs more frequently in covered stents [51]. Stent fracture, cholangitis, bleeding, intestinal perforation are other described complications [51].

### 3.2. EUS-Guided Gastro-Enteral Anastomosis

Endoscopic ultrasonography-guided gastrojejunostomy (EUS-GJ) has been described as an available alternative for the management of GOO [12]. EUS-GJ implies the transgastric identification of the jejunum distal to the obstruction and subsequent creation of a gastro-enteric anastomosis [60]. Technically, the anastomosis is performed with the placement of a lumen-apposing metal stent (LAMS), a bi-flagged-covered stent with uncovered ends, to reduce the risk for migration (Figure 5) [61].

The reported technical and clinical success rates of EUS-GJ were, respectively, 92% and 90%; however, the procedure is burdened by the risk of complications such as stent dislodgement, peritonitis, bleeding, pneumoperitoneum, hemoperitoneum, abdominal pain, and leakage [62]. Similar results were reported by McCarty et al. in a metanalysis of 5 studies, including 199 patients treated with EUS-GJ [63]. They reported a technical success rate of 92.90%, a clinical success rate of 90.11%, an AEs occurrence rate of 5.61%, and a reintervention rate of 11.43%. Boghossian et al. performed a metanalysis in order to compare the clinical outcomes of EUS-GJ, endoscopic stents (ES), and surgical gastrojejunostomy (SGJ) [64]. Comparing EUS-GJ and ES, the technical success rate was 93% and 98%, respectively; the clinical success rate was 88% and 78%, respectively; the AEs rate was 11% and 31%, respectively; and the occurrence of stent occlusion was 3% and 24%, respectively. Moreover, comparing EUS-GJ and SGJ, the technical success rate was 91% and 100%, respectively; the clinical success rate was 86% and 90%, respectively; the AEs rate was 11% and 10%, respectively; and the length of hospital stay was longer in SGJ. Perez-Miranda et al., in a multicentric study, compared EUS-GJ and SGJ and found no differences in terms of technical and clinical success rate between the two groups; however, SGJ was reported to have a significantly higher incidence of AEs when compared with EUS-GJ (41% vs. 12%) [65].

## 4. Caeliac Plexus Neurolisis

Right upper quadrant or epigastric pain is one of the most common symptoms associated with advanced GBC occurring in more than 50% of patients [66]. In this setting, the pain can be determined by the gallbladder overdistention or the malignant infiltration of the surrounding tissues [67]. As for other cancers, the medical approach to pain follows a stepwise approach from NSAIDs to strong opioids [68]. However, opioids are burdened by several complications, including constipation, nausea, vomiting, sedation, tolerance, and dependence [69]. Interventional treatments such as nerve blocks, neuromodulation, intrathecal drug delivery, and radiotherapy have been described as alternative or adjuvant therapy to classical pain drugs [70]. Fluoroscopy, ultrasound, and computed tomography-guided nerve block and neurolysis are often used to treat incoercible pain. The upper abdominal organs are innerved by the celiac plexus, located retroperitoneally around the origin of the celiac artery [71]. Celiac plexus neurolysis (CPN) implies the chemical ablation of nerve tissues to reduce the transmission of pain sensation. Formerly, CPN was performed surgically or with radiologic guidance. However, the celiac plexus is easily accessed by EUS, and consequently, an EUS-guided CPN has been developed [72]. The EUS-CPN procedure involves the EUS identification of the celiac plexus, the insertion of a needle within the plexus, and the injection of chemical substances (e.g., alcohol). Alternatively, an EUS-guided celiac plexus RFA ablation can be performed [73]. EUS-guided CPN was reported as a safe technique with minor adverse events such as hypotension and diarrhea [13].

The role of EUS-CPN has been widely evaluated in the management of pancreatic cancer-related pain, with a reported efficacy in alleviating abdominal pain in 72% of cases [74]. Wyse J.M. performed a randomized double-blinded controlled trial and randomly divided 96 patients presenting with pancreatic cancer-related pain into two groups: 48 were treated with EUS-CPN, and 48 were treated with standard medical treatments [75]. Pain relief at 1 and 3 months follow-up was greater in the EUS-CPN group, and morphine consumption was lower at 3 months follow-up in the EUS-CPN group. The role of EUS-CPN in the management of pain related to GBC has been less widely investigated. Praveer et al. published the largest case series of EUS-CPN performed in 21 patients with GBC [13]. They documented a technical success rate of 90%. Pain relief was experienced in 95% at week 2, in 63% and 61% at week 4 and week 8; there was a significant reduction in analgesics consumption at week 2 and 4.

## 5. Locoregional Therapies

Considering the poor prognosis and the lack of curative treatment for advanced GBC, new endoscopic locoregional treatments have been described to prolong the patient’s survival and improve the quality of life. Endoscopic radiofrequency ablation (RFA), intraluminal brachytherapy (IBT), and endoscopic photodynamic therapy (PDT) have been studied for the management of biliary malignancies [76,77,78,79].

RFA involves the insertion of a bipolar electrode within the tumor tissue and the application of an alternating electric current. The production of high temperatures leads to tissue-coagulative necrosis and consequently reduction in tumor size [80]. The intrabiliary application of RFA has been described for malignant biliary stenosis [81]. Endobiliary-RFA was demonstrated to prolong the biliary stent patency and the patients’ overall survival [77]. Lee et al. reported the use of endobiliary-RFA in 30 patients with malignant biliary stricture (among them, 2 were diagnosed with GBC) [82]. The overall duration of stent patency and survival were, respectively, 236 days and 383 days. Post-procedural adverse events occurred in 10% of patients (two pancreatitis and one cholangitis).

IBT provides the local delivery of high-dose radiation, enhancing the anti-neoplastic effect inside the tumor tissue and reducing the exposure to radiation of adjacent structures. The endoscopic approach involves the placement of a naso-biliary tube (NBT) in the biliary tree, and the insertion of a BT catheter inside the NBT for the delivery of high-dose radiation or the placement of a ^124^I seeds stent through the biliary stricture [78,83]. Xu et al., in a metanalysis of 12 studies involving 641 patients with malignant biliary obstruction (among them 45 were diagnosed with GBC), reported that patients treated with IBT were associated with lower incidence of stent occlusion and longer overall survival when compared with patients treated with biliary stent alone [84]. No differences in terms of complications were reported.

PDT involves the intravenous injection of a photosensitizer followed by irradiation with a specific wavelength of light in order to produce a selective ablation of cancer cells and a modulation of the cancer microenvironment [85]. Several studies analyzed the role of PTD in biliary malignancy, particularly in cholangiocarcinoma [86]. Leggett et al., in a meta-analysis of six studies, demonstrated that the association of biliary stenting with PDT was associated with improved biliary drainage, better quality of life, and longer survival [87]. However, the role of PDT in advanced GBC has not been evaluated yet, and more studies are needed to assess its safety and efficacy.

## 6. Conclusions

The overall prognosis of GBC is poor, with an estimated 5-year survival of 5% to 13% [8]. The only curative approach is still surgery today; however, more than 50% of patients present at the diagnosis had an advanced GBC not amenable to surgical treatments [45]. Jaundice, cholangitis, vomiting, abdominal pain may be the first clinical presentation of GBC requiring a palliative treatment. Endoscopy is rising as the gold-standard mini-invasive technique for the management of palliation in several oncological settings. In patients presenting with advanced GBC associated with cholangitis or jaundice, endoscopic biliary drainage is frequently the preferred procedure [20]. Cholangitis is a life-threatening condition and is considered an absolute indication for biliary drainage [19]. Jaundice associated with pruritus or abdominal pain is also considered an indication for biliary drainage in order to improve the quality of life and prolong the patient’s overall survival and permit treatments by chemo-radiation therapy [19]. ERCP is the master technique to perform endoscopic biliary drainage [32]. With palliative purpose, the uncovered metallic stents (U-SEMS) are generally preferred as their longer patency and lower risk for stent migration. A complete endoscopic biliary drainage (>50% of liver volume) should always be performed in order to reduce the risk for supra-infection and liver failure [32]. When ERCP is not feasible (e.g., for altered anatomy or failed cannulation for tight stricture), the EUS-BD can be considered a valid alternative [39]. The choice among the several EUS procedures available (e.g., choledochoduodenostomy, choledochogastrostomy, hepaticoduodenostomy, and hepaticogastrostomy) depend on local facilities and should be carefully evaluated in each singular patient [42]. GOO is another possible complication of advanced GBC associated with a strong reduction in patients’ quality of life [46]. Endoscopy offers a mini-invasive alternative to surgical gastrojejunostomy [52]. The endoscopic approach includes the placement of an endoluminal SEMS or the performance of a EUS-GJ. Endoluminal SEMS are the most used in the daily clinical practice; however, the interest on EUS-GJ is rising as this procedure appears to be safe and effective [62]. EUS offers also the opportunity to manage incoercible pain in patients with advanced GBC. Therefore, EUS-guided celiac plexus neurolysis seems to be an effective and safe procedure [72]. However, its role has been widely investigated mainly for pancreatic cancer and few data are available on advanced GBC. Finally, the role of locoregional treatments such as RFA, IBT, and PDT has been described in GBC [76,77,78]. However, their efficacy has not been deeply investigated yet and more studied are needed to use these procedures in daily practice.

In conclusion, endoscopic palliation in advanced GBC plays a crucial role in the improvement of patients’ quality of life and the prolongation of overall survival. Each patient should be carefully examined in order to choose the endoscopic technique that better fits with the specific clinical case. The different endoscopic techniques, surgery, and interventional radiology should be combined in order to obtain the best clinical outcomes.

## Figures and Tables

**Figure 1 cancers-14-01686-f001:**
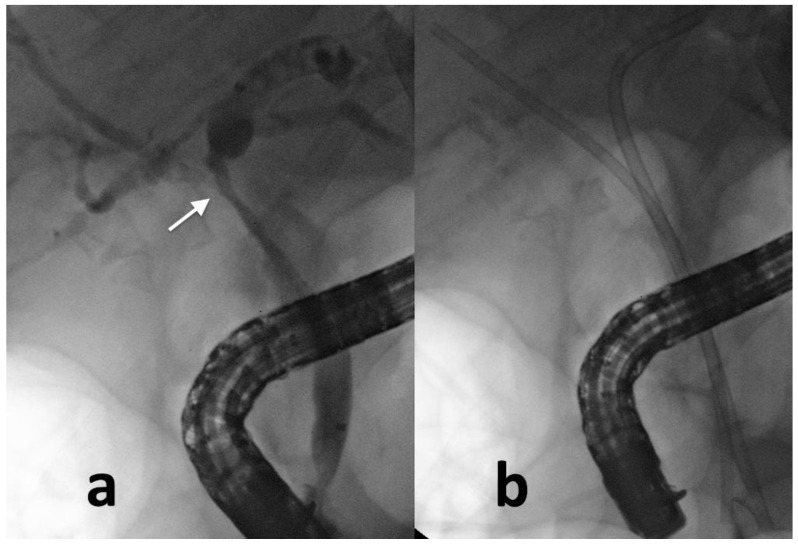
Gallbladder cancer invading the main hepatic confluence (arrow) (**a**); two plastic stents (**b**) are inserted to drain the bile ducts and to obtain jaundice relief.

**Figure 2 cancers-14-01686-f002:**
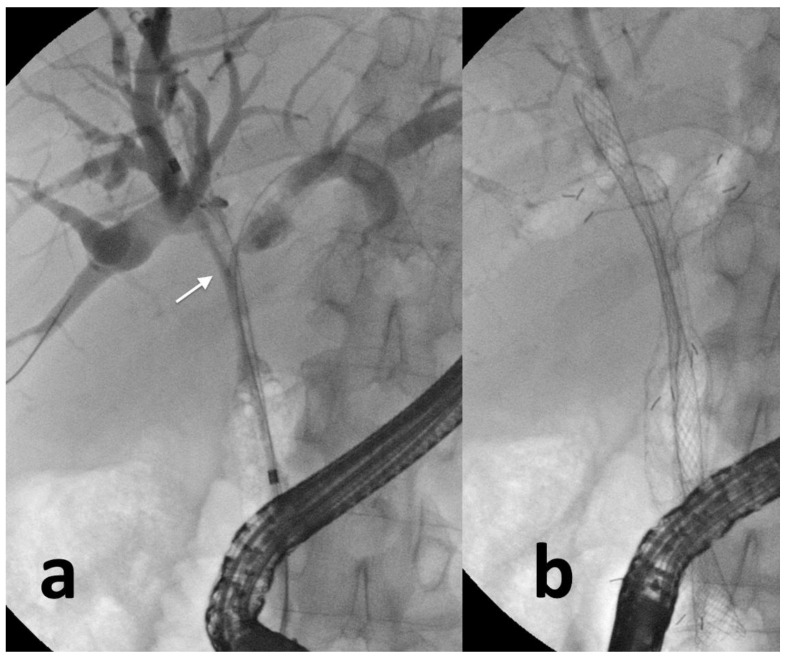
Complex hilar biliary stricture (arrow) in a patient with gallbladder cancer (**a**); three self-expandable metal stents are inserted (**b**) to obtain definitive palliation.

**Figure 3 cancers-14-01686-f003:**
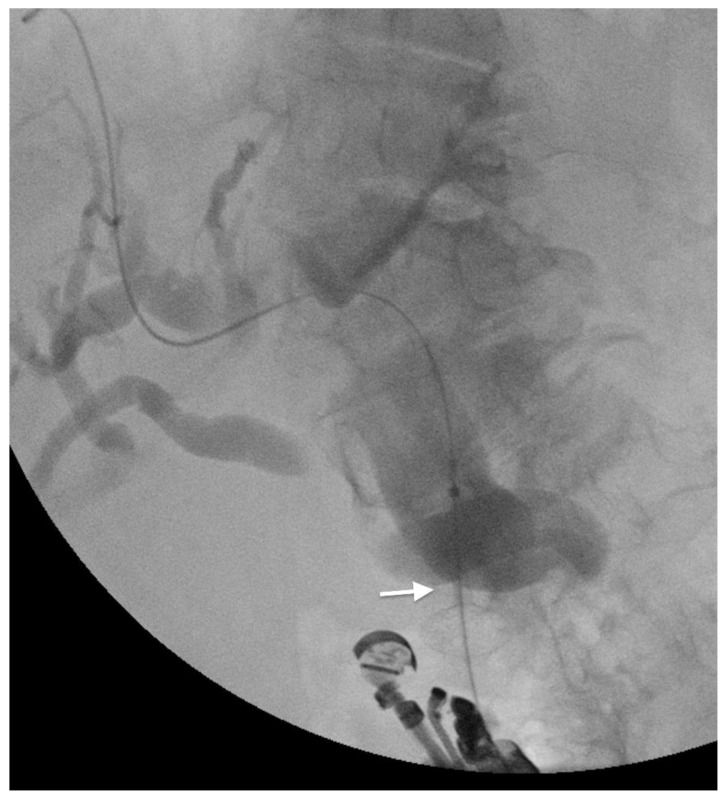
EUS-guided choledoco-duodenostomy with a lumen-apposing metal stent (arrow) to drain the bile duct in a patient with gallbladder cancer.

**Figure 4 cancers-14-01686-f004:**
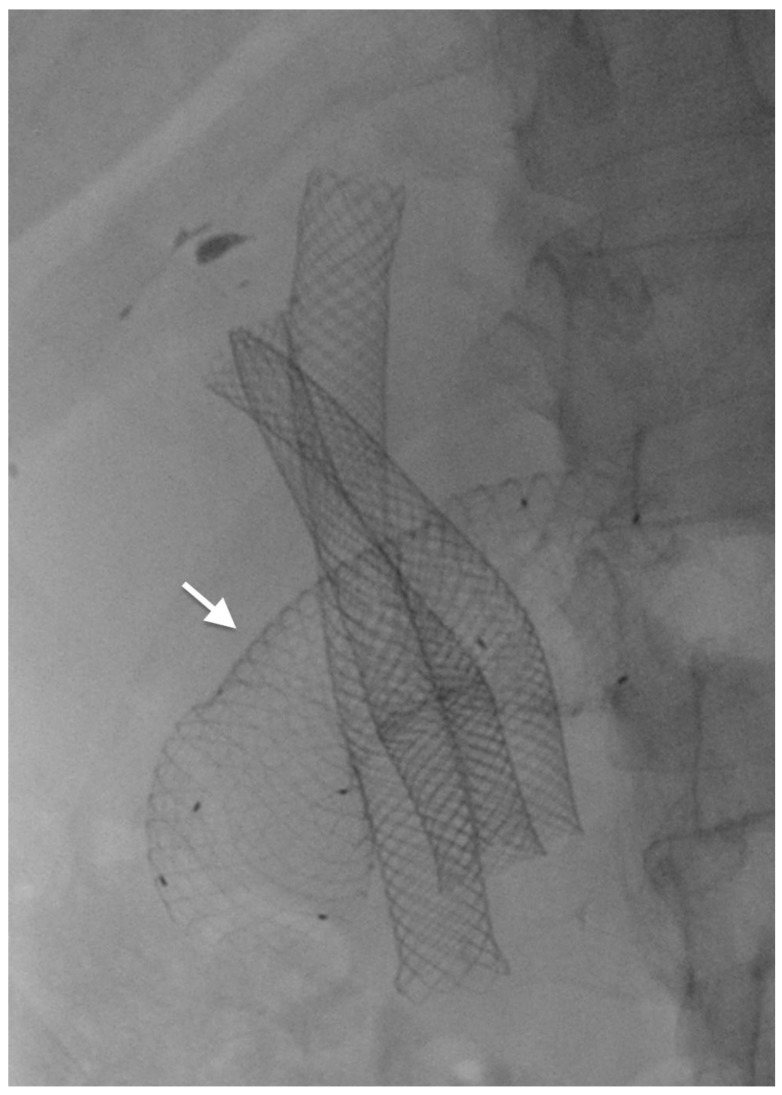
A duodenal uncovered metal stent (arrow) is placed to palliate a duodenal stricture secondary to invasion from gallbladder cancer. Three metal biliary stents were previously placed for jaundice palliation.

**Figure 5 cancers-14-01686-f005:**
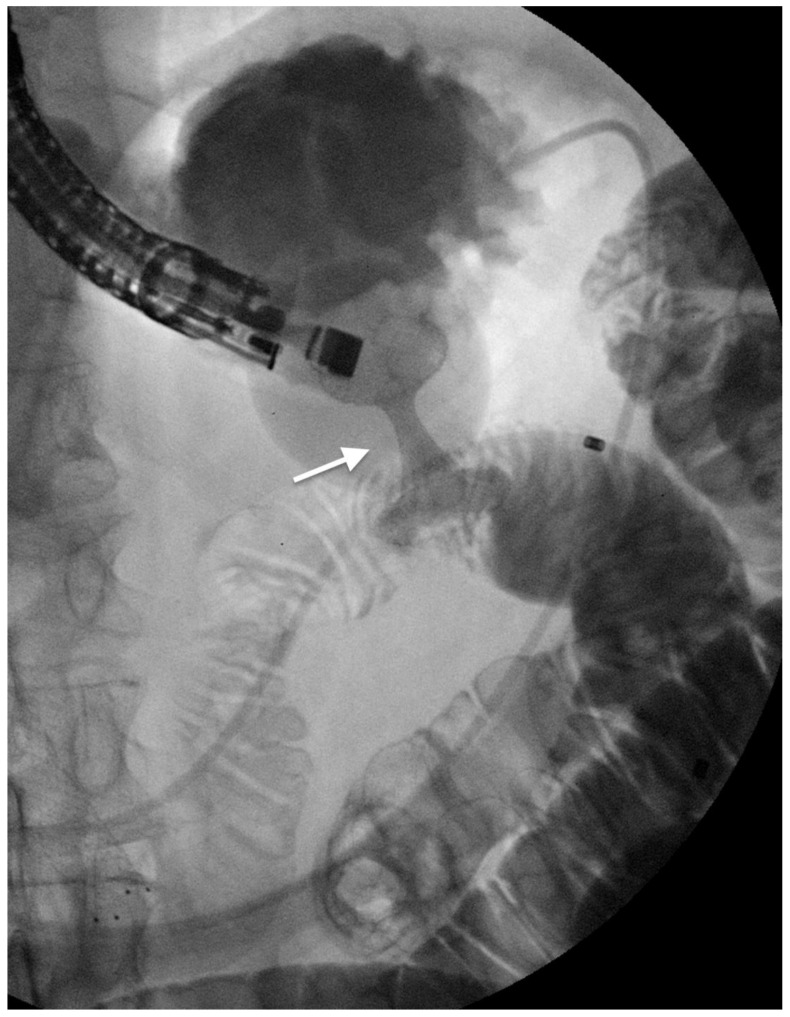
EUS-guided gastro-jejunostomy by lumen-apposing metal stent (arrow).

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
