# Peer review of "Palliation in Gallbladder Cancer: The Role of Gastrointestinal Endoscopy"

_cancers, 2022, doi:10.3390/cancers14071686_

Round 1

Reviewer 1 Report

The manuscript is an excellent and interesting

review with wonderful figures. It was a pleasure to 

read.

Author Response

We are pleased to know that the reviewer appreciated our study. 

Reviewer 2 Report

This manuscript is a review that introduced latest knowledge in palliative treatment for gallbladder cancer using endoscopic technique. The authors described endoscopic treatment in palliation for cancer-associated symptoms including obstructive jaundice, gastric outlet obstruction and abdominal pain. Generally, it is a well-written paper which includes informative information.  The results will be of interest to clinicians in the field.

However, the following minor issues require clarification:

Minor

  1. I recommend that the author describe how to conduct literature searches.
  2. (P6L201) Please replace “ERCP” with “EBD”.
  3. (P7L225-7) The authors should discuss the indication of covered metallic stents in patients with malignant GOO.
  4. Please provide the data regarding adverse events in EUS-CPN.
  5. The conclusion section is too long. Especially, the first paragraph should be more summarized. Or it might be unnecessary.
  6. I found some mis-spelling words. Please check and correct them.

Author Response

1. We aimed to performe a narrative review, as we didn't provide a "methods" section. 

2. We replaced "ERCP" with "EBD" as requested. 

3. The indication for endoscopic treatment of GOO are explained in chapter 3 "Gastric Outlet Obstruction". 

4. Thank you for the suggestion, we added data on adverse events in EUS-guided CPN. 

5. Thank you for the suggestion, we summurized the first part of the conclusions as requested. 

Reviewer 3 Report

The manuscript reviews the palliative therapeutic approaches that can be performed through endoscopy in patients with billiary carcinoma. As the authors point out, these patients are most frequently diagnosed at an advanced stage with limited curative possibilities. Therefore, palliation is an important part of patient care and endoscopic means are often less invasive and offer significant benefits for the quality of life. The authors provide a review of current endoscopic procedures that can be performed as palliative measures in patients with advanced billiary carcinoma. The manuscript is well structured and well written, summarizing the essentials for each procedure in part, with advantages compared to other procedures and with potential complications and outcomes. Only minor typographical errors. However, these procedures are well known and have been in current use for a long time. 

Author Response

We kindly thank the reviewer for the comment provided.  We checked and corrrected the typographical errors as requested.